# A Dynamic Numerical Simulation on the Grouting Timing in Retained Rib of Pillarless Mining

Xianyang Yu [1,2,*], Jinhao Xie [1,2], Yanju Wu [3], Qiuhong Wu [1,2], Zizheng Zhang [1,2] and Hai Wu [1,2]

1    Hunan Provincial Key Laboratory of Safe Mining Techniques of Coal Mines, Hunan University of Science and Technology, Xiangtan 411201, China
2    Work Safety Key Lab on Prevention and Control of Gas and Roof Disasters for Southern Coal Mines, Hunan University of Science and Technology, Xiangtan 411201, China
3    Luoyang CITIC Imaging Intelligent Technology Co., Ltd., Luoyang 471000, China
*    Correspondence: 1010092@hnust.edu.cn

**Abstract:** A dynamic numerical method is established to simulate the stability of the surrounding rocks of the retained roadway in FLAC3D, along with a double-yield constitutive model to simulate the re-compaction process of gangue and a strain-softening constitutive model to simulate the strain-softening characteristic of the coal and the grouted fragmented coal after yielding. The simulation reveals that the grouting slurry diffusion range, the mining affecting the stage behind the working face and the retained coal rib deformation are closely interrelated. Under severe mining-induced stress, the integrity of the surrounding rock is more likely to be destroyed, accompanied by a large number of cracks developing and gradually expanding in the surrounding rocks. The roadway deformation increases in a rapid manner. Meanwhile, the grouting diffusing range increases gradually. The simulation conducted in this study indicates that the optimum support effect can be achieved by grouting in the section before and after the working face affected by the high mining-induced stress. A timely grouting can be used to construct an enhanced surrounding rock-bolting-grouting support system and maintain the stability of the retained roadway.

**Keywords:** grouting timing; numerical simulation; strain-softening constitutive model; pillarless mining





## 1. Introduction

The retained roadways often experience various types of serious problems with large surrounding rock deformation in pillarless mining [1–5]. A number of field cases show that the deformation of retained coal ribs is one of main causes behind the rib-to-rib convergence. During this process, the retained rib with loose and fragmented coal becomes the weakest segment of the roadway with a limited bearing capacity [6–9], which often undergoes severe deformations, and may even affect the stability of the entire roadway [10,11].

Grouting is considered as an effective support method to reinforce the fragmented coal and restore the load-bearing capacity of the retained coal rib. Under the long-term impact of the mining-induced stress, the fragmented zone in the retained coal rib gradually develops and expands deep into the coal rib. Along with this process, the shallow coal permeability improves while the roadway surrounding rock stability gradually decreases [12–14]. For the retained roadway in pillarless mining, the slurry diffusion range and the integrity of the coal rib are dependent on each other in a contradictory and unified manner. In order to maintain the stability of the roadway, a balance must be achieved between the two aspects.

FLAC3D is an effective tool suitable for simulating the nonlinear and large deformation behavior of geotechnical materials and their supporting structures, especially for simulating the plastic flow of materials after yielding, which has been widely used in the stability analysis of the surrounding rock of the retained roadway [15,16]. In this study, the FLAC3D was adopted based on multiple constitutive models such as the double yield, plastic softening, Mohr–Coulomb and corresponding algorithms to construct a dynamic numerical

method to simulate the stability of retained roadway surrounding rock. In addition, the plastic zone development in the roadway during the process of roadway deformation and the optimal grouting timing of the fragmented coal rib were studied.

The study of rock mass control primarily focuses on understanding the movement patterns of coal and rock layers influenced by mining activities. As the coal is extracted from the longwall mining face, the surrounding rock layers shift and fill the void left behind by the coal extraction. This process results in a series of intense movements, including the movement of rock layers between the coal seam roof and the surface, leading to surface subsidence [17,18]; the development of support pressure on both sides of the working face; and the convergence of the goaf and the roof/floor of the working face [19,20]. The basic principles of the dynamic model for coal mining proposed in this study can be applied to various aspects of ground control, such as roadway deformation, roof stability and surface subsidence above the coal mining face. Other researchers in this field can use this dynamic model in combination with the actual geological conditions of their projects to conduct research on ground control.

## 2. The Principle of the Dynamic Numerical Simulation

### 2.1. Simulation Flow Chart

After the coal is mined, caved zones, fractured zones and continuous bending zones tend to be formed above the goaf. In longwall mining, the gangue falls to fill the goaf and subsequently is compressed. In this process, the stress concentration at the coal rib is also gradually weakened. Roof caving is a complex process involving the rock mass deformation, separation and destruction [21]. Therefore, different criteria should be adopted between the gangue falling and the rock yielding in numerical simulation. Rock yielding takes place when the rock stress exceeds its strength, while the roof caving indicates the state of fragmented rock blocks falling into the goaf due to their own weight. Singh [22] simulated the pressure of the hydraulic support on the long wall working face. In this study, two criteria of the gangue falling were adopted including the rock shear strain reaching 0.25 or the roof rock subsiding more than 1 m. The immediate roof caving is mainly caused by the separation of the rock layer, which is ultimately destroyed by the tensile deformation. In the process of numerical simulation, roof subsidence and mining height showed a positive correlation. As a result, maximum plastic tensile strain can be used as the criterion for determining whether the roof surrounding rock is located in the caved zone [23]. The element tensile strain in FLAC3D is defined as Equation (1) [24]:

$$\varepsilon^t = \varepsilon^e + \varepsilon^p \tag{1}$$

where $\varepsilon^t$ is the element tensile strain, $\varepsilon^e$ refers to the element elastic tensile strain and $\varepsilon^p$ indicates the element plastic strain.

The rock experiences tensile failure $\varepsilon^p >> \varepsilon^e$ at the time when $\varepsilon^t \approx \varepsilon^p$. In this study, $\varepsilon^p > 0.05$ was used as the criterion to evaluate the roof surrounding rock located in the caved zone [25].

Another important issue about the goaf is the caved zone height. The gangue expansion ratio can be used to determine the height of the caved zone [25]. Studies have shown that the expansion ratio of gangue ($K_P$) is 1.1–1.5 [26], with a falling gangue porosity of about 0.3 [27,28].

Assuming that the gangue fills the mined-out space, the falling gangue porosity is given by:

$$\frac{h}{h + h_m} = 0.3 \tag{2}$$

where $h$ is mining height, representing the voids in the falling gangue, and $h + h_m$ represents the total volume of the falling gangue.

So, the expansion ratio can be expressed as Equation (3):

$$K_P = \frac{h + h_m}{h_m} \tag{3}$$

where $h$ is mining height and $h_m$ refers to the height of caved zone. When $K_P \leq 1.43$, the goaf has been filled.

The calculating process of the numerical simulation model is demonstrated in Figure 1. During the simulation, some routine checks were performed every 100 steps after the calculation initiates. When the equilibrium state is reached, the next 10 m will be excavated. Before the model reaches equilibrium, with the roof and floor contacting, the next 10 m will be excavated also. In other cases, the next 100 steps are expected to be iterated, leading to the next preformation. The calculation will be terminated when all coal seams are mined and an equilibrium is reached. The gangue in the caved zone is assigned to the double-yield constitutive model with the strength of the surrounding rock in the fractured zone reduced.

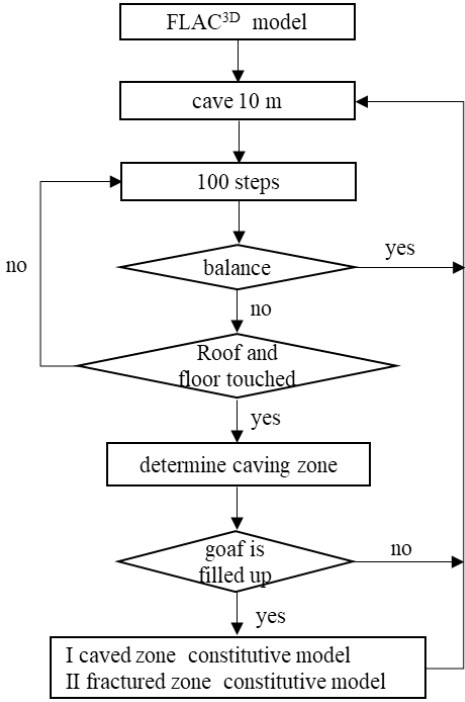

**Figure 1.** The principle of dynamic numerical simulation on pillarless mining.

*2.2. Surrounding Rock Constitutive Model*

2.2.1. The Constitutive Model of Caved Zone

Gangue, as a stack of rock blocks with compressible properties, contains a large number of pores. The double-yield constitutive model in FLAC3D takes both the shear-tension yield and the permanent volume reduction caused by the compression of the material into account. The double-yield constitutive model can well simulate the re-compaction process of gangue [29]. The yield criterion is shown in Equation (4) [24]:

$$\begin{cases} f^s = \sigma_1 - \sigma_3 N_\varphi + 2c\sqrt{N_\varphi} \\ f^t = \sigma^t - \sigma_3 \\ f^v = \frac{1}{3}(\sigma_1 + \sigma_2 + \sigma_3) + \mathrm{p}_c \end{cases} \tag{4}$$

where $N_\varphi = \frac{1+\sin\varphi}{1-\sin\varphi}$, $\varphi$ is internal friction angle, $c$ refers to the cohesion, $\sigma^t$ indicates the tensile strength and $\mathrm{p}_c$ stands for the cap pressure.

Generally, the gangue cohesion is 0 MPa with an internal friction angle of 30° [30,31]. The Salamon stress–strain equation can be used to invert the cap pressure parameters [32], as shown in Equation (5):

$$\sigma = \frac{E_0\varepsilon_v}{1 - \varepsilon_v/\varepsilon_v^m} \tag{5}$$

where $E_0$ is the initial secant modulus, $\varepsilon_v$ refers to the volumetric strain and $\varepsilon_v^m$ indicates the maximum volumetric strain. With the gangue expansion ratio, $\varepsilon_v^m$ can be obtained, as shown in Equation (6) [33]:

$$\varepsilon_v^m = \frac{K_p - 1}{K_p} \tag{6}$$

$E_0$ can be obtained through laboratory tests. The existing research indicates that the initial secant elastic modulus of gangue is about 80 MPa [33,34].

### 2.2.2. Constitutive Model of Coal and Fragmented Coal

In the Mohr–Coulomb model, it is assumed that the cohesion, internal friction angle, tensile strength and shear expansion property will not change after yielding. In fact, previous studies have shown that coal shows a significant strain-softening characteristic after yielding [35,36]. The strain-softening constitutive model can better simulate the roadway surrounding rock stability with the high mining-induced stress in deep mines. Users can define the cohesion, internal friction angle, tensile strength and shear expansion property after the peak of the coal. The yield criterion is the same as the Mohr–Coulomb model [24,37]:

$$\begin{cases} f^s = \sigma_1 - \sigma_3 N_\varphi + 2c\sqrt{N_\varphi} \\ f^t = \sigma^t - \sigma_3 \end{cases} \tag{7}$$

The reference of each variable in the equation remains unchanged through this study.

The failure process of coal under pressure shows the characteristics of strain softening, which was manifested by the decrease in cohesion and internal friction angle [38]. For the roadway surrounding rock in a fragmented rheological state, under the circumstance that the supplementary support is not provided in time, the roadway tends to soon enter into a state of instability. Grouting can cement the fractured surrounding rock into a whole body with the bearing capacity, which is equivalent to improving the cohesion and internal friction angle of the surrounding rock. The increased ratio of the two is determined by the rock type, failure state, grouting material [39] and grouting parameters. Existing studies suggest that the lower surrounding rock strength and the higher the fracture degree, the more the generalized rock strength will increase after grouted. At the water–cement ratio of 0.8, the strength of the grouting geocomposite can be restored to 31.43%, 57.11% and 85.51% of the strength of intact coal after being grouted with P.O42.5 Portland cement, ultra-fine cement and MasterRoc MP 364, which have 2.55, 2.15 and 5.28 times the residual strength of fragmented coal, respectively [40].

In the FLAC3D, the strain-softening constitutive model allows users to express the internal friction angle, cohesion, and dilatancy angle of the frail material as a piecewise function of shear plastic strain [24], as shown in Figure 2. $\varepsilon^{ps}$ represents the damage degree of the rock behind the peak. With the increase in $\varepsilon^{ps}$, the strength of the roadway surrounding rock gradually decreases.

The piecewise function of the cohesion and internal friction angle of the surrounding rock versus the shear plastic strain can be used in FLAC3D to simulate the improvement of the overall strength of the fractured surrounding rock by grouting. For the low-strength surrounding rocks, the post-peak strain softening mainly shows a decrease in cohesion. From the peak strength to the residual strength, the cohesion is reduced by about 50%, while the internal friction angle basically stays unchanged [33]. According to the research of Lu and Liu [38,41], the coal strain-softening parameters in the numerical simulation are shown in Figure 2.

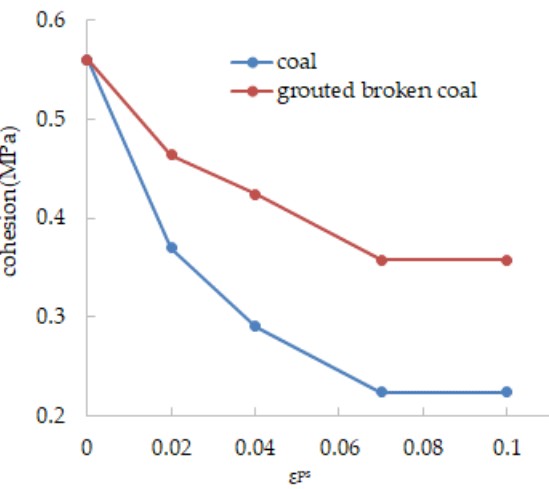

**Figure 2.** Strain-softening constitutive model of fragmented coal and grouted fragmented coal.

### 2.2.3. Constitutive Model of Fractured Zone and Immediate Floor

After the immediate roof falling, the rock in the fractured zone is in a state of decompression. In the process of re-compaction of the gangue, the stress of the surrounding rock in the fracture zone is restored. In this process, the rock layer tends to experience the unloading–loading cyclic stress leading to the possibility of destroying the integrity of the surrounding rock and a post-peak fracture state. Both the elastic modulus and strength of the rock will decrease. In order to simulate this case, the strength reduction method was used to treat the rock in the fracture zone [42], and the strength proprieties of rock in the fractured zone above the goaf were reduced after mining. Similar to the fractured zone, the immediate floor undergoes a large-scale pressure relief process after mining, resulting in lower integrity and strength. The fractured zone and the immediate floor were simulated via the Mohr–Coulomb model, and the yield criterion of the two is shown in Equation (7).

### 3. Simulation Model

This model was based on the Number 1111(1) Panel of Zhuji Mine, and the tailgate of this panel was retained after mining. The 1111(1) working face has a width of 260 m and a length of 1700 m, making it relatively large in size. In order to reduce the scale of the numerical calculation model and decrease computational complexity, this study adopts a symmetric approach by modeling only half of the working face in the strike width direction. Additionally, a model length of 200 m is used by truncating the model in the length direction. As shown in Figure 3, a three-dimensional numerical model was established based on the actual coal seam occurrence condition and the properties of roof and floor surrounding rocks. The model dimensions were 200 m length × 168 m width × 100 m height. The thickness of the coal seam and the mining height were 2.0 m, the size of the roadway cross section was 5.0 m wide × 3.0 m high and the filling wall width was 3.0 m.

The model includes a total of 744,980 nodes and 701,400 blocks. In order to improve the simulation accuracy, the coal seam, immediate roof and immediate floor grids are appropriately densified with a block size of 0.5 m length × 1.0 m width × 0.5 m height. A pressure of 16.8 MPa is applied to the upper boundary of the numerical simulation model, with displacement constraints applied to the other five faces. The model meshing is shown in Figure 4.

The rock physical and mechanical parameters used in the numerical simulation are shown in Table 1.

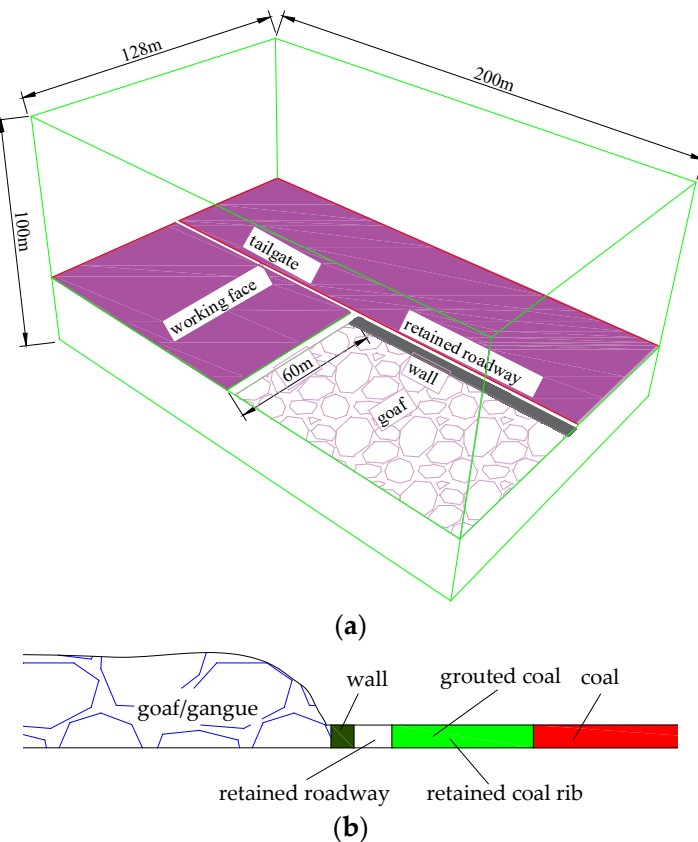

(**a**)

(**b**)

**Figure 3.** The retained roadway simulation model in FLAC3D. (**a**) Simulation model. (**b**) Cross-sectional diagram of the retained roadway section in the model.

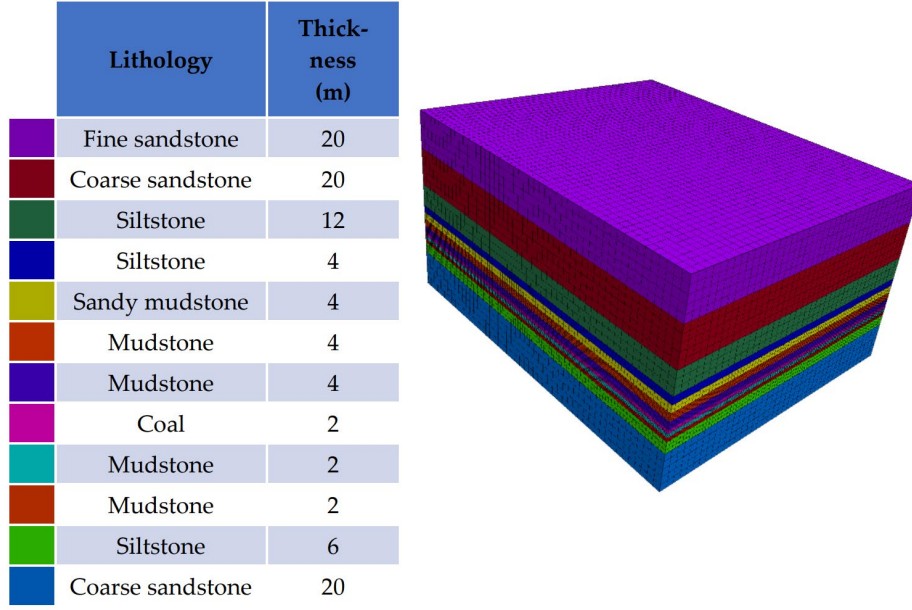

| Lithology | Thickness (m) |
|---|---|
| Fine sandstone | 20 |
| Coarse sandstone | 20 |
| Siltstone | 12 |
| Siltstone | 4 |
| Sandy mudstone | 4 |
| Mudstone | 4 |
| Mudstone | 4 |
| Coal | 2 |
| Mudstone | 2 |
| Mudstone | 2 |
| Siltstone | 6 |
| Coarse sandstone | 20 |

**Figure 4.** Model meshing in FLAC3D.

**Table 1.** Rock physical and mechanical parameters.

| Thickness (m) | Lithology | Density (kg·m⁻³) | Bulk Modulus (GPa) | Shear Modulus (GPa) | Friction (°) | Cohesion (MPa) | Tensile Strength (MPa) |
|---|---|---|---|---|---|---|---|
| 20 | Fine sandstone | 2750 | 12.74 | 12.35 | 39 | 6.15 | 2.51 |
| 20 | Coarse sandstone | 2750 | 10.21 | 10.25 | 38 | 5.58 | 2.31 |
| 12 | Siltstone | 2700 | 10.35 | 8.74 | 38 | 4.15 | 2.24 |
| 4 | Siltstone | 2600 | 8.35 | 5.74 | 36 | 3.15 | 2.01 |
| 4 | Sandy mudstone | 2500 | 3.6 | 2.89 | 29 | 2.35 | 1.32 |
| 4 | Mudstone | 2000 | 2.88 | 1.53 | 26 | 1.07 | 0.98 |
| 4 | Mudstone | 1700 | 2.51 | 1.58 | 25 | 0.95 | 0.82 |
| 2 | Coal | 1400 | 1.87 | 0.63 | 21 | 0.56 | 0.05 |
| 2 | Mudstone | 2000 | 2.64 | 1.54 | 23 | 1.45 | 1.07 |
| 2 | Mudstone | 2000 | 2.88 | 1.73 | 24 | 1.67 | 1.21 |
| 6 | Siltstone | 2400 | 3.88 | 3.53 | 26 | 2.17 | 1.54 |
| 20 | Coarse sandstone | 2700 | 7.35 | 7.74 | 36 | 3.15 | 1.69 |
| - | Filling wall | 2500 | 1.72 | 0.86 | 25 | 0.81 | 2.35 |
| - | Caved zone [33] | 1700 | 13.89 | 0.15 | 30 | 0.001 | 0 |

## 4. Plastic Zone and Grouting Diffusing Range Development

As shown in Figure 5, when the working face advances by 120 m, the plastic zone begins to develop in the coal seam.

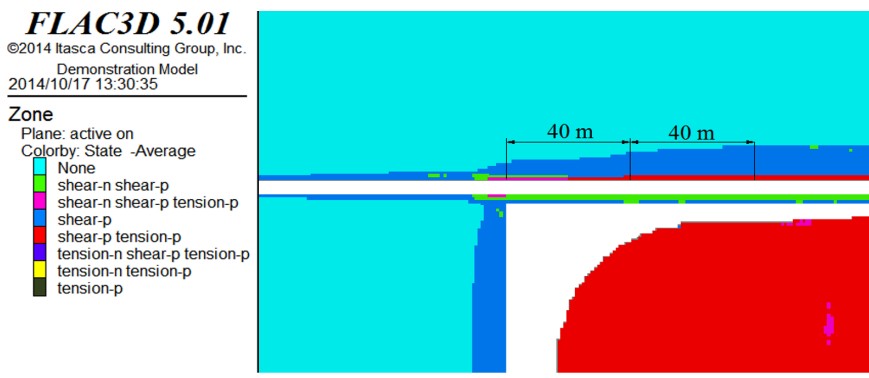

**Figure 5.** Plastic zone development in coal rib.

Outside the mining disturbed area, the depth of the plastic zone of the roadway was only about 1 m, mainly manifested as shear failure. Cracks were mostly developed from the original small cracks caused by the mining-induced stress during the roadway excavation and weathering of coal on the roadway surface. The original bolting support maintained the stability of the roadway in the driving stage.

Outside the mining disturbance range, the depth of the plastic zone in the surrounding rock of the roadway was only about 1 m. The fragmented surrounding rock was mainly in the form of shear failure. The main cause of cracks was the development of original small cracks caused by the mining stress during the excavation of the roadway and the weathering of the coal on the roadway surface. During the excavation process, the bolt support maintained the stability of the roadway before mining.

With the advancement of the working face, cracks gradually penetrated into the retained coal rib. As the coal rib displacement increased, some tensile failures were observed in the roadway surrounding rock. In the area of 10 m depth before the working face, the plastic zone began to expand under the influence of mining-induced stress. The mining-induced stress caused a large number of new cracks in the retained coal rib and caused the original cracks to expand further. Since the roadway adopted the active strong bolting support, and the main roof above the roadway was not fragmented or rotated and subsided, the range of the plastic zone in coal rib had not been greatly expanded at this time.

During the advancing process of the working face, the cracks gradually penetrated into the retained coal rib. With the increase in the displacement of the coal rib, a tensile failure area appeared on the surface of the surrounding rock of the roadway. In the area 10 m ahead

of the working face, the plastic zone began to expand under the action of mining stress. The mining stress caused a large number of new cracks to be generated in the retained coal rib, and at the same time caused the original cracks to further expand. Because the roadway was supported by active strong bolts, the main roof was not fragmented, rotated or subsided. During this process, the plastic zone of the coal rib did not expand significantly.

The expansion of the plastic zone in the coal rib behind the working face shows the same stage characteristics as the deformation of the retained roadway, which can be divided into 0–40 m mining-affected severe section, 40–80 m mining-affected moderate section and out by 80 m mining-affected stable section behind the working face.

The depth of the plastic zone in the severely mining-affected section increased sharply, from 6.0 m depth near the working face to 10 m depth at 30 m behind the working face. After the main roof breaking, rotating, and subsiding, the stress concentration occurred in the retained rib, which caused the shallow part of the coal rib to be crushed and squeezed out. At the same time, the weakening and failure of the bolting supporting structure also occurred. At this point, the surrounding rock-supporting load-bearing structure was already in a post-peak failure state. The roadway surrounding rock needed supplementary support in time to avoid a slow and continuous plastic flow in the future.

In the severe mining affected section, the depth of the plastic zone increased from 6.0 m near the working face to 10 m at 30 m behind the working face. During the fracture, rotation and subsidence of the main roof, stress increased and intensified at the retained coal rib, resulting in the crushing and extrusion of the shallow part of the coal rib. This process was accompanied by the weakening and failure of the anchor support structure. At this time, the surrounding rock-support load-bearing structure was in a post-peak failure state. The surrounding rock of the roadway needed timely supplementary support to prevent slow and continuous flow deformation in the future.

In the mining-affected stable section, the plastic zone in the retained coal rib basically no longer expanded and remained at 12 m. However, it is worth noting that the retained coal rib deformation reached 750 mm at this time, and the roadway will quickly enter a state of rheological instability if timely and effective supplementary support is not installed.

The roadway surrounding rock is a low-strength, layered rock mass, embedded with many original joints. Under the influence of mining stress [43], many cracks tend to develop in the shallow part of surrounding rock. Grouting in roadway surrounding rocks is generally conducted under a low pressure, which is often maintained at 2 MPa. During low-pressure grouting, the slurry spreads mainly along the cracks in the rock, and the development of the cracks determines the surrounding rock permeability.

According to studies by Niu [44], a shear plastic strain reaching 0.007 is used as the criterion for the breaking state of the surrounding rock, and the zone where $\varepsilon^{ps} > 0.007$ is taken as the range to initiate the slurry injection. Figure 6 shows the ranges that can be grouted at different positions before and behind the working face, which is shown in the color green in the figure.

A comparison of the slurry diffusion range, the mining affecting stage behind working face and the retained coal rib deformation revels that the three elements are closely interrelated, as shown in Figure 7. Before the working face, the deformation of the roadway outside the mining-affected zone is small, and the depth of the plastic zone in the surrounding rock is only about 1 m. The surrounding rock remains intact with the limited shear plastic strain in the plastic zone and a small grouting diffusing range. Under the influence of severe mining-induced stress, the integrity of the surrounding rock was damaged before and after the work, and a large number of cracks were generated and gradually expand in the surrounding rock. The roadway deformation increased rapidly. In addition, the shear flow characteristics are observed in the damaged surrounding rock. In the simulation, the rapidly increasing amount of shear plastic strain in the surrounding rock gradually increased the grouting diffusing range. The roof activity tends to be stable while the stress environment of the roadway surrounding rock reaches a balanced state in the mining-affected stable section. At this stage, a large-scale plastic failure occurred in the

shallow part of roadway surrounding rock, which the grouting diffusing range would no longer increase.

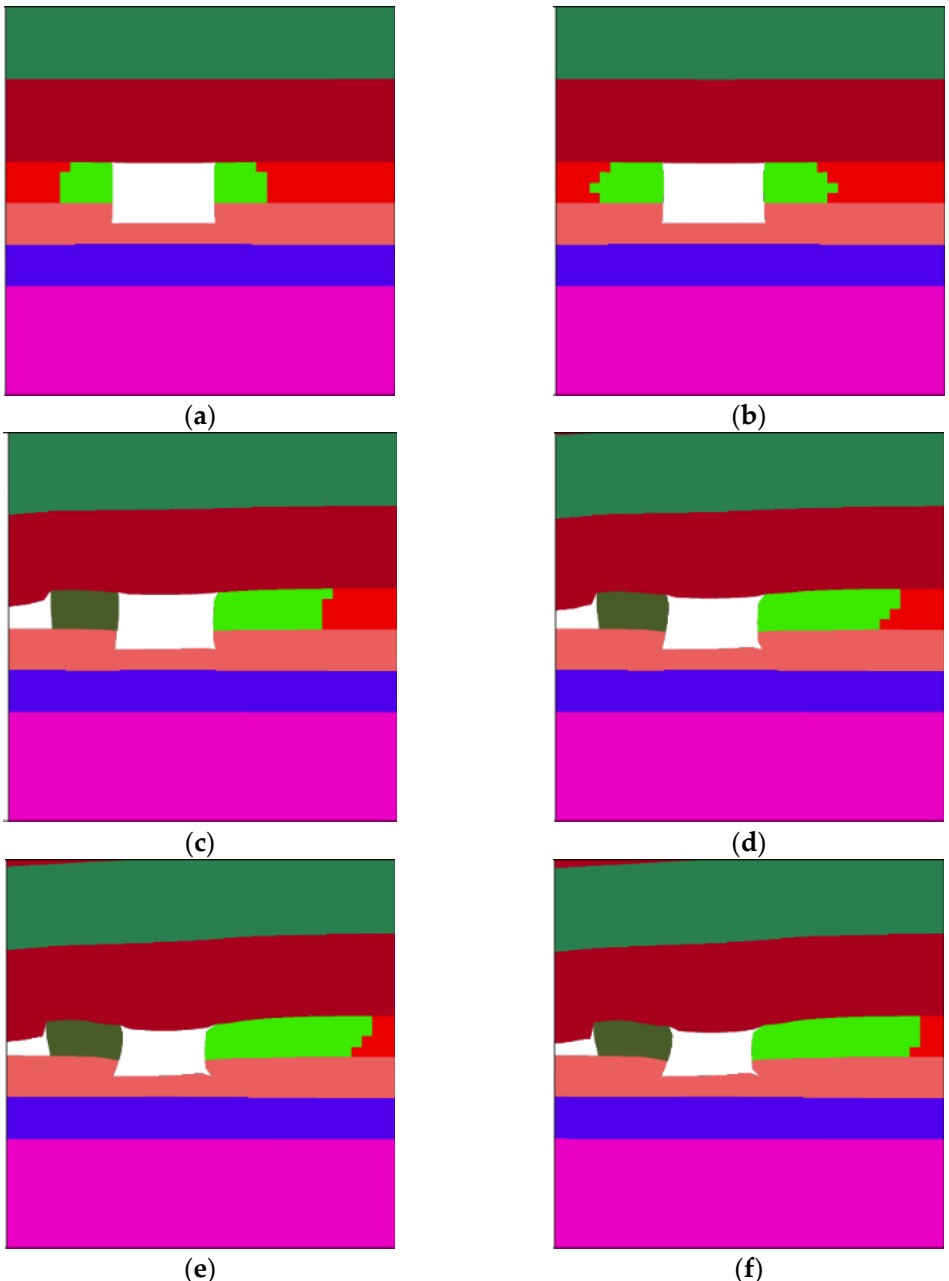

**Figure 6.** Grouting diffusing range development during mining. (**a**) 30 m before working face. (**b**) 10 m before working face. (**c**) 10 m behind working face. (**d**) 30 m behind working face. (**e**) 60 m behind working face. (**f**) 100 m behind working face. Note: ■ represents the range where the slurry can be injected into coal rib; ■ represents the wall of retained roadway.

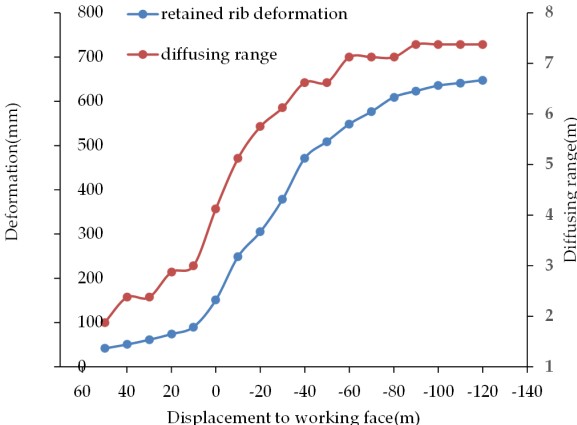

**Figure 7.** Relationship between diffusing range, rib deformation and grouting site.

## 5. Discussion

The timing of grouting is one of the most important parameters for grouting in mining roadways. Timely grouting in the surrounding rock can repair the damaged bolt structure in time. The roadway support structure and the surrounding rock can be effectively coupled and become an effective bearing structure again [45,46].

Grouting in retained coal rib was carried out at different distances from the working face in this simulation. If grouting is performed at different sites before and after the working face, Figure 8 shows the deformation of the retained coal rib when the grouting position is 120 m behind the working face. On the whole, whenever grouting was used to consolidate the fragmented surrounding rocks, the deformation of the retained coal rib was effectively controlled to varying degrees, as shown in Figure 8. The curve in Figure 8 presents a "U" shape, indicating an optimal timing of grouting in the retained coal rib of pillarless mining.

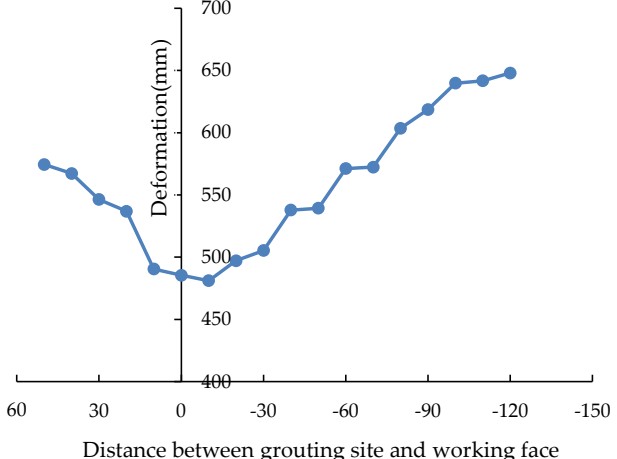

**Figure 8.** Retained coal rib deformation grouted at different distances from working face.

The principle of optimal grouting timing in the retained coal rib of pillarless mining is shown in Figure 9. The solid line 1 in the figure represents the deformation of the retained coal rib without grouting reinforcement, and the dashed lines 2–5 represent the deformation of the retained coal rib after the grouting is carried out at different sites before and after the working face.

As illustrated by solid line 1, the roadway experiences some large deformations and even instability without the grouting. In this case, the surrounding rock of the roadway is in a loose and fragmented state, accompanied with a complete loss of the anchoring force. The roadway is in a plastic flow state under the low support force.

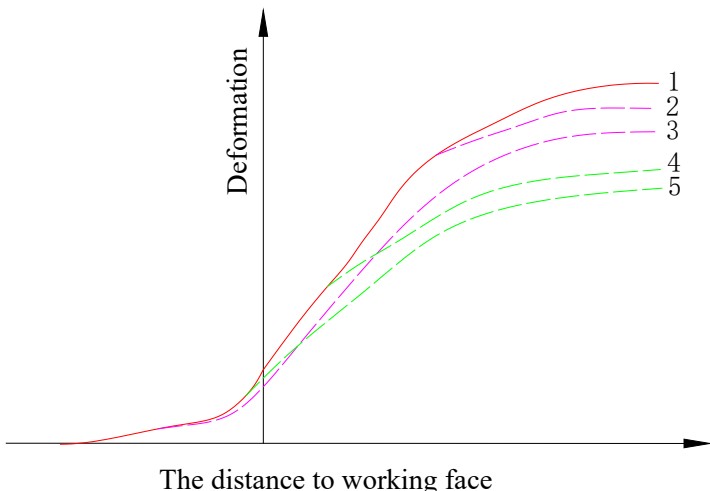

**Figure 9.** Principle of optimal grouting timing in retained coal rib. Note: 1—no grouting; 2—grouting too late; 3—grouting too early; and 4 and 5—optimal grouting timing.

The dashed line 2 represents the deformation of the retained coal rib when grouting was added at a later than optimal timing. In the mining-affected stable section, the main roof above the roadway forms a stable voussoir beam structure, with a low mining-induced stress in the surrounding rock. In addition, the deformation speed was also low. However, the fragmented range expanded beyond the anchoring range, and the roadway surrounding rock enters a state of slow and continuous plastic flow.

The dashed line 3 represents the deformation of the retained coal rib when the grouting was added at an earlier than optimal timing. When not affected by mining-induced stress, the fissures in the retained rib are original joints and small cracks caused by roadway excavation. The crack apertures were small and the connectivity degree was low.

The dashed lines 4 and 5 are representative points of the optimal grouting timing. The best support effect can be obtained by grouting at the sections affected by high mining-induced stress before and behind the working face. According to the study by YU [33], the observation of the bore scope could be used to study the development of cracks in the retained coal rib and optimize the grouting time and reasonable reinforcement methods.

Under the influence of mining-induced stress, many cracks appear in the coal rib, leading to the continuous expansion of cracks and the original joints in the coal seam.

It should be noted that the fractured zone of the coal rib is still within the anchoring range. In this case, the interaction system between the bolt structure and surrounding rock is in a critically stable state, and the bolt system exerts its maximum supporting effect, as shown in Figure 10 [40].

In the minding sites, the fast-hardening sulfoaluminate cement is a commonly used grouting cement, whose uniaxial compressive strength can reach 30 MPa in 1 day and can be increased to 42.5 MPa in 3 days. The recommended method of support is to use a grout with a water–cement ratio of 0.8, grouting 20–30 m before the working face [40].

Grouting can cement the fragmented rock mass into a whole body, relieve the stress concentration between the fragmented rock blocks, improve the whole mechanical properties of the fragmented surrounding rock, and facilitate the fragmented surrounding rock to re-form a bearing structure. Meanwhile, the grouting can repair the failed bolting structure, which greatly increases the supporting efficiency of the bolts in large-deformation roadways. More importantly, a timely grouting can lead to a surrounding rock-bolting-grouting support system with active support capabilities and maintain the stability of the retained roadway.

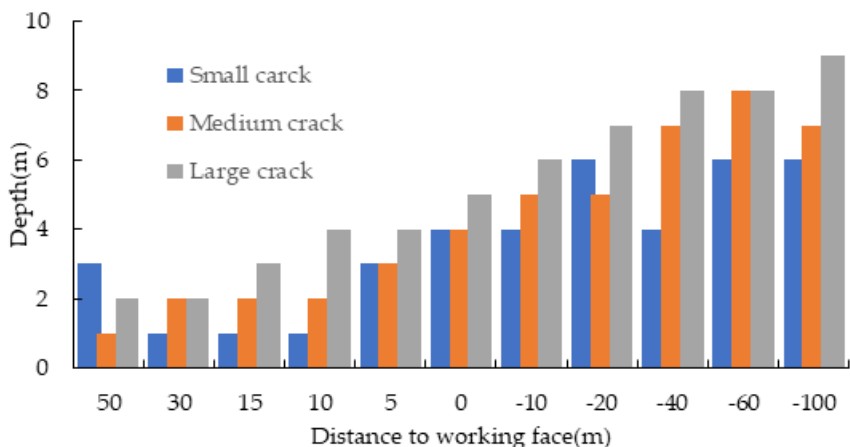

**Figure 10.** Statistical graph of crack development in boreholes around working face.

## 6. Conclusions

According to multiple simulations conducted in this study, the following conclusions were drawn.

(1) A dynamic numerical method was established and adopted to simulate the surrounding rock stability of the retained roadway in FLAC3D.

(2) In this study, the double-yield constitutive model can be used effectively to simulate the compaction process after the gangue falls into the goaf while the strain-softening constitutive model can effectively simulate the strain-softening behavior of coal and grouted coal blocks after peaks in FLAC3D.

(3) The grouting slurry diffusion range, the mining affecting stage behind the working face and the retained coal rib deformation are closely interrelated. Severe mining-induced stresses lead to the destruction of the integrity of the surrounding rock of the roadway. During this process, many cracks are produced in the surrounding rock and the cracks gradually expand. The roadway deformation increases rapidly. At the same time, the grouting diffusing range increases gradually.

(4) An optimal timing in the grouting process can be identified to maximize the grouting efficiency in the retained coal rib of pillarless mining. The optimum support effect can be achieved by grouting in the sections before and after the working face where the roadway surrounding rock is affected by high mining-induced stress. A timely grouting can lead to a surrounding rock-bolting-grouting support system with active support capabilities and maintain the stability of the retained roadway.

**Author Contributions:** Conceptualization, X.Y.; methodology, Y.W. and Q.W.; software, J.X.; validation, Y.W.; formal analysis, Y.W. and Z.Z.; investigation, H.W.; resources, J.X.; data curation, Z.Z.; writing—original draft preparation, X.Y.; writing—review and editing, X.Y. and J.X.; visualization, H.W.; supervision, X.Y.; project administration, X.Y.; and funding acquisition, X.Y. All authors have read and agreed to the published version of the manuscript.

**Funding:** This research was funded by The National Science Foundation of China, grant number 51904102 and Open Foundation of Hunan Provincial Key Laboratory of Safe Mining Techniques of Coal Mines, grant number E21733 and Scientific Research Fund of Hunan Provincial Education Department, grant number 21C0353.

**Institutional Review Board Statement:** Not applicable.

**Informed Consent Statement:** Not applicable.

**Data Availability Statement:** The data in this article are available from the corresponding author.

**Conflicts of Interest:** The authors declare no conflict of interest.

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
