# Peer review of "A Dynamic Numerical Simulation on the Grouting Timing in Retained Rib of Pillarless Mining"

_applsci, doi:10.3390/app13169479_

Round 1

Reviewer 1 Report

This paper deals with the analysis of grouting timing in retained rib of pillarless mining using dynamic numerical simulation and takes a fundamental approach for investigation.

The topic of the paper is important and requires investigation. The writing is good with relevant review of the current state of knowledge. Although the literature review could be improved (e.g., add a couple of more references related to previous similar work) and more critical discussion be added, it is in general acceptable. 

The figures and illustrations are relevant and appropriate. The data presented are useful.

This is amazing work. Thank you for your contributions.

I suggest publication as is.

Author Response

We appreciate your positive feedback and recommendation for publication as is. Thank you for your valuable input.

Reviewer 2 Report

The manuscript is not sound enough from the text description, the theory, and the figures.

1. There are a lot of terminologies: retained roadway, gangue, coal and grouted broken coal, retained coal rib, goaf (the authors have misspelled it as golf in line 59) filling, roof caving, gangue falling. Please use some sketches to explain these terminologies; if possible, these specific terminologies should be reduced.

2. Why do you only consider tensile failure, while at the same time, the Mohr-Coulomb model is also adopted? I think the M-C model is better for describing the shear-compression failure (especially when cohesion = 0 for gangue).

3. I cannot find any logic among different mathematical expressions. How do you come up with Equations (4) and (5)? How do you incorporate these equations into your double-yield constitutive model (3)? Furthermore, from (2) and (5), epsilon_v^m = h/(h + h_m); why is that?

4. The legend in Figure 6 is too small. Again, it would be best if you tried to enrich this figure by adding some descriptions to tell the reviewer the physical meaning of these different colors. How do you connect Figure 6 with Figure 1?

5. Figure 1 could also be enhanced, for example, what is the "balance"?

6. Related reference that also mentions rock separation, mining-induced cracks, grouting, and stress: http://koreascience.or.kr/article/JAKO202111236745354.page (DOI: 10.12989/GAE.2021.24.5.479), 10.1021/acsomega.0c01626, 10.1016/j.engfailanal.2022.106762

English writing must be enhanced. The transition from sentence to sentence is not clear. For example, line 75 - line 77: how do you know that the value of K_p is 1.43? The scientific logic is missing. Line 15: "the mining affecting stage behind working face and the retained coal rib deformation", it seems that this phrase contains two different mechanisms that are closely related, but the original text failed to express this relationship.

Reviewer 3 Report

Dear Authors,

I have some remarks:

1/ please correct the font type in the list of authors

2/ please extend the introduction to deformations caused by underground mining in the upper parts of the rock mass and on the surface of the terrain

3/ please add foreign literature, e.g. European, concerning the discussed problem and deformations (e.g. from Poland: prof. Strzalkowski, prof. Scigala, prof. Szafulera, prof. Orwat)

4/ in the simulation, the width of the working face seems too small compared to the actual operating conditions

5/ the simulation should be done for different geometries of the mining excavation -> universality of results

6/ Figure 10 - what do the gray bars mean? Add it to the legend

7/ please add doi to some literature items

8/ are the conlusions universal? Can they be used for other operations, in other regions? please add this information

nothing

Round 2

Reviewer 2 Report

Fine.